# SASSL: Enhancing Self-Supervised Learning via Neural Style Transfer

**Renan A. Rojas-Gomez**[*]                    *renanar2@illinois.edu*
*University of Illinois at Urbana-Champaign*

**Karan Singhal**                              *karansinghal@google.com*
*Google Research*

**Ali Etemad**                                 *ali.etemad@queensu.ca*
*Queen's University, Canada*

**Alex Bijamov**                               *abijamov@google.com*
*Google DeepMind*

**Warren R. Morningstar**                      *wmorning@google.com*
*Google DeepMind*

**Philip Andrew Mansfield**                    *memes@google.com*
*Google DeepMind*

[*]*Work done during an internship at Google.*

**Reviewed on OpenReview:** *https://openreview.net/forum?id=NxhXtkPYsk*

## Abstract

Existing data augmentation in self-supervised learning, while diverse, fails to preserve the inherent structure of natural images. This results in distorted augmented samples with compromised semantic information, ultimately impacting downstream performance. To overcome this limitation, we propose *SASSL: Style Augmentations for Self Supervised Learning*, a novel data augmentation technique based on Neural Style Transfer. SASSL decouples semantic and stylistic attributes in images and applies transformations exclusively to their style while preserving content, generating diverse samples that better retain semantic information. SASSL boosts top-1 image classification accuracy on ImageNet by up to 2 percentage points compared to established self-supervised methods like MoCo, SimCLR, and BYOL, while achieving superior transfer learning performance across various datasets. Because SASSL can be performed asynchronously as part of the data augmentation pipeline, these performance impacts can be obtained with no change in pretraining throughput.

## 1 Introduction

Data labelling is a challenging and expensive process, which often serves as a barrier to build machine learning models to solve real-world problems. Self-supervised learning (SSL) is an emerging machine learning paradigm that helps to alleviate the challenges of data labelling, by using large corpora of unlabeled data to pretrain models to learn robust and general representations. These representations can be efficiently transferred to downstream tasks, resulting in performant models which can be constructed without access to large pools of labeled data. SSL methods have shown promising results in recent years, matching and in some cases exceeding the performance of bespoke supervised models with small amounts of labelled data.

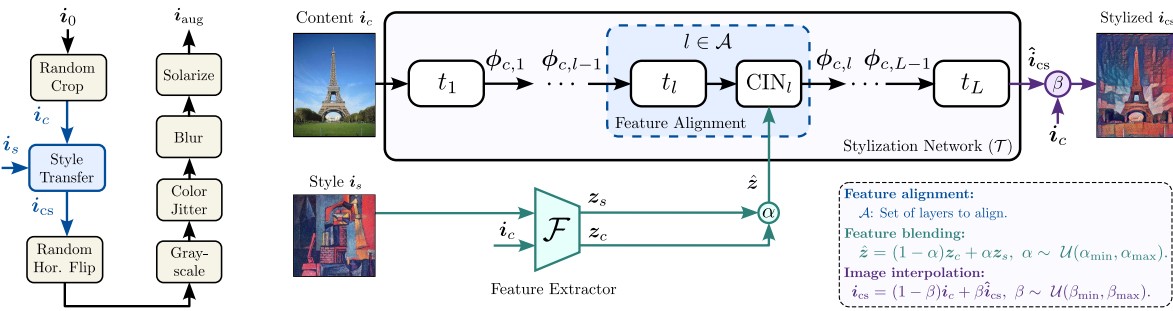

(a) **Data augmentation pipeline.**    (b) **Style Transfer preprocessing.**

Figure 1: **Towards diverse SSL data augmentation via Neural Style Transfer.** We propose *SASSL*, a novel augmentation technique that leverages Style Transfer to create pretraining views that are semantically aware, focusing solely on modifying the image's appearance while preserving its content. SASSL combines the image's content with the texture of an external reference style, generating augmented views that better retain the image's semantic meaning. By incorporating Style Transfer into traditional SSL augmentation pipelines and controlling the stylization strength through gradual blending of style features and pixel values, SASSL promotes stronger representations compared to well-established SSL methods.

Given the lack of labels, SSL relies on pretext tasks, *i.e.*, predefined tasks where pseudo-labels can be generated. These include contrastive learning (Chen et al., 2020a; He et al., 2020), clustering (Caron et al., 2021; 2020; Assran et al., 2022), and generative modeling (He et al., 2022; Devlin et al., 2018). Many pretext tasks involve training the model to distinguish between different views of the same input and inputs corresponding to different samples. For these tasks, the way input data is augmented is crucial to learn useful invariances and extract robust representations (Chen et al., 2020a). While state-of-the-art augmentations incorporate a wide range of color, spectral and spatial transformations, they often disregard the natural structure of an image. As a result, SSL pretraining methods may generate augmented samples with degraded semantic information, and may be less able to capture diverse visual attributes.

To tackle this challenge, we propose *Style Augmentations for Self Supervised Learning (SASSL)*, a novel SSL data augmentation technique based on Neural Style Transfer to generate semantically consistent augmented samples. In contrast to augmentation techniques operating on specific formats (e.g. pixel or spectral domain), SASSL disentangles an image into perceptual (*style*) and semantic (*content*) representations that are learned from data. Applying transformations only to the style of an image while preserving its content, we can generate images with diverse appearance that retain the original semantic properties.

**Our contributions:**

- We propose SASSL, a novel data augmentation technique based on Style Transfer that naturally preserves semantic properties while diversifying style (Section 4).
- We empirically show improved downstream performance on ImageNet Deng et al. (2009) by incorporating SASSL in methods such as MoCo, BYOL and SimCLR without hyperparameter tuning (Sections 5.1, 5.4).
- We show SASSL learns stronger representations by measuring their transfer learning capabilities on various datasets. Our method boosts linear probing performance by up to 10% and fine-tuning by up to 6% on out-of-distribution tasks (Section 5.2).

# 2    Related Work

## 2.1    Data Augmentation in SSL

Typical data augmentation methods applied to vision tasks include image cropping and resizing, flipping, rotation, color augmentation, noise addition, and solarization. Examples of methods using these are MoCo (He et al., 2020), SimCLR (Chen et al., 2020a), BYOL (Grill et al., 2020), and SimSiam (Chen & He, 2021), among others. Other work (Caron et al., 2020) shows how generating additional augmentations using this strategy can improve performance relative to two view approaches, though this strategy decreases throughput

and requires additional compute. Other research explore how to select augmentation hyperparameters to learn more robust and general features (Wagner et al., 2022; Reed et al., 2021; Tian et al., 2020). In contrast, SASSL proposes a new preprocessing technique that can be incorporated in existing augmentation pipelines, boosting performance without additional hyperparameter tuning or changes in pretraining throughput.

Previous SSL work has explored semantic-aware augmentation approaches. Purushwalkam & Gupta (2020) leverage natural video transformations occuring in videos as an alternative to learning from object-centric datasets. Lee et al. (2021) introduces an auxiliary loss to capture the difference between augmented views, leading to better performance on tasks where semantic information is lost due to aggressive augmentation. Bai et al. (2022) propose an alternative augmentation pipeline to prevent loss of semantics by gradually increasing the strength of augmentations. While these methods modify the pretraining loss and require keeping track of augmentation hyperparameters, SASSL integrates seamlessly into existing pipelines without additional loss terms or auxiliary data. Our method complements default data augmentation pipelines with a content-preserving transformation to obtain stronger image representations.

## 2.2 Neural Style Transfer

Recent image generation and Style Transfer algorithms (Liu et al., 2021; Heitz et al., 2021; Jing et al., 2020; Yoo et al., 2019; Risser et al., 2017; Gatys et al., 2017; Chen et al., 2021a) use CNNs to measure the texture similarity between two images in terms of the distance between their neural activations. Feature maps of a pretrained classifier such as VGG-19 (Simonyan & Zisserman, 2014) are extracted and low-order moments of their distribution (Kessy et al., 2018; Huang & Belongie, 2017; Sheng et al., 2018) are used as texture descriptors. By matching such feature statistics, these techniques have shown promising results transferring texture between arbitrary images, improving over classic texture synthesis approaches (Portilla & Simoncelli, 2000; Zhu et al., 2000; Heeger & Bergen, 1995).

A large body of work focuses on *artistic* applications, reproducing an artwork style over a scene of interest. These methods adopt either (i) an *iterative optimization* approach (Risser et al., 2017; Gatys et al., 2016; 2017; Li et al., 2017a), where an initial guess is gradually transformed to depict a style of interest or (ii) an *autoencoding* approach (Liu et al., 2021; Li et al., 2017b; Chen et al., 2021a; Wang et al., 2020), where one or more CNN image generators are trained to impose a target texture in a single forward pass. While selecting an approach implies a trade-off between synthesis quality and computational cost, in both cases the generated stylization shows an unnatural appearance, *i.e.*, it often lacks the qualities of a real-world scene.

In the context of data augmentation for supervised learning, Geirhos et al. (2018) and Zheng et al. (2019) addressed texture bias and generated training samples via pre-stylized datasets or stylizing using a small collection of style images. Hong et al. (2021) explored Style Transfer to improve robustness against adversarial attacks. Jackson et al. (2019) incorporated Style Transfer as a transformation in the augmentation pipeline. While their approach of randomly mixing content and style representations yield promising results, it neglects potential distortions introduced by the Style Transfer network bottleneck. SASSL, in contrast, integrates Style Transfer in a self-supervised setting. Our approach generates diverse augmentations via either pre-computed style representations from external datasets or in-batch stylization with training samples as style references. Importantly, SASSL preserves semantic information through pixel interpolation and feature blending, mitigating the loss of details inherent in Style Transfer networks.

## 3 Preliminaries

### 3.1 Self-Supervised Learning

Traditional SSL methods learn compressed representations by maximizing the agreement between differently augmented views of the same data example in a latent space. They do this following the template originally proposed by SimCLR (Chen et al., 2020a). In this setup the input is split into multiple views using data augmentations, encoded into a representation, and then further projected into an embedding over which the loss is computed. There are many potential augmentations that can be used, including (but not limited to) random cropping, flipping, color jitter, blurring, and solarization. By maximizing the similarity of

augmented training samples, the network learns to create robust representations that separate meaningful semantic content from simple distortions that could occur in the real world, and which should not affect the semantic content of an image.

Given a batch of $N$ input images $\{\boldsymbol{i}_k\}_{k=1}^N$, $2N$ augmented samples are generated by applying distinct transformations to each image. These transformations correspond to the same data augmentation pipeline. Let $R$ correspond to all possible augmentations. Then, positive pairs correspond to augmented views of the same input sample, and negative pairs correspond to views coming from different input images. Based on this, the $2N$ augmented samples $\{\tilde{\boldsymbol{i}}_l\}_{l=1}^{2N}$ can be organized so that indices $l = 2k - 1$ and $l = 2k$ correspond to views of the $k$-th input sample

$$\tilde{\boldsymbol{i}}_{2k-1} = r(\boldsymbol{i}_k), \quad \tilde{\boldsymbol{i}}_{2k} = \hat{r}(\boldsymbol{i}_k), \quad \hat{r}, r \sim R \tag{1}$$

Once augmented views are obtained, a representation is computed using an image encoder (typically a CNN model). The representations are then fed to a projection head which further compresses them into a lower-dimensional manifold where different views of the same image are close together and those from different images are far apart. Let $h$ and $g$ be the encoder (e.g. a ResNet-50 backbone) and projection head (e.g. an MLP layer), respectively. Then, embeddings are obtained for each augmented sample as $\boldsymbol{z}_l = g \circ h(\tilde{\boldsymbol{i}}_l)$.

SimCLR uses the normalized temperature-scaled cross entropy loss (NT-Xent) to learn how to identify positive pairs of augmented samples. First, the cosine similarity of every pair of embeddings is computed

$$s_{m,n} = \frac{\langle \boldsymbol{z}_m, \boldsymbol{z}_n \rangle}{\|\boldsymbol{z}_m\|\|\boldsymbol{z}_n\|} \tag{2}$$

The model is then trained using a contrastive loss by comparing the embeddings of positives, forcing them to be similar to each other. Since the loss is normalized, it naturally forces the representations of views from two different images (negatives) to be distant from each other.

$$\mathcal{L} = \frac{1}{2N} \sum_{k=1}^N \left[ \ell(2k-1, 2k) + \ell(2k, 2k-1) \right], \quad \ell(m, n) = -\log\left( \frac{\exp(s_{m,n}/\tau)}{\sum_{l=1}^{2N} \mathbb{1}_{m \neq n} \exp(s_{m,l}/\tau)} \right) \tag{3}$$

where $\tau \in \mathbb{R}_{++}$ is the temperature factor and $\mathbb{1}$ the indicator function. While SimCLR is a simple framework, it pushed the state-of-the-art significantly on a wide range of downstream tasks including image classification, object detection, and semantic segmentation.

Follow up works to SimCLR such as MoCo (Chen et al., 2020b; 2021b), BYOL (Grill et al., 2020) and SimSiam (Chen & He, 2021), among others (Caron et al., 2020; 2021; Assran et al., 2022; Zbontar et al., 2021; Bardes et al., 2021), have largely maintained this template, but have proposed modifications to this setup (e.g. new losses, architectures, or augmentation strategies) which attempt to further improve the downstream task performance.

### 3.2 Neural Style Transfer

Style Transfer techniques combine the semantics (*content*) of an image with the visual characteristics (*style*) of another image. These assume that the statistics of shallower layers of a trained CNN encode style, while deeper layers encode content. Seminal techniques are based on an optimization-based approach, passing a pair of content and style images to a CNN encoder and optimizing over a randomly initialized image to produce activations with similar statistics to the style image at shallower layers and similar activations to the content image at deeper ones (Gatys et al., 2015). This way, a *stylized* image is generated, comprising the semantic and texture attributes of interest.

While optimization-based methods generate a diverse stylization due to a random image initialization, autoencoding methods utilize an image decoder to efficiently stylize arbitrary image pairs on a single forward pass. In what follows, we introduce the autoencoding Style Transfer technique adopted by our proposed method due on its generalization and efficiency properties. For an in-depth survey of Style Transfer methods, refer to Jing et al. (2019).

**Fast Style Transfer.** Dumoulin et al. (2017) proposed an arbitrary Style Transfer method with remarkable generalization properties. Their algorithm, *Fast Style Transfer*, accurately represents unseen artistic styles by training a model to predict first and second moments of latent image representations at multiple scales. Such moments are used as arguments of a special form of instance normalization, denominated *conditional instance normalization* (CIN), to impose style over arbitrary input images.

Given a content image $\boldsymbol{i}_c \in \mathbb{R}^{C \times H_c \times W_c}$ and a style image $\boldsymbol{i}_s \in \mathbb{R}^{C \times H_s \times W_s}$, Fast Style Transfer produces a stylized image $\boldsymbol{i}_{\text{cs}}$ that corresponds to

$$\boldsymbol{i}_{\text{cs}} = \mathcal{T}(\boldsymbol{i}_c, \boldsymbol{z}_s) \in \mathbb{R}^{C \times H_c \times W_c} \tag{4}$$

where $\mathcal{T}$ is a stylization network and $\boldsymbol{z}_s = \mathcal{F}(\boldsymbol{i}_s) \in \mathbb{R}^D$ is an embedding extracted from the style image via a feature extractor $\mathcal{F}$, e.g., InceptionV3 (Szegedy et al., 2016).

We assume $\boldsymbol{z}_s$ to be a contracted embedding of the style image ($D \ll CH_sW_s$). The stylization network $\mathcal{T}$ is comprised by $L$ blocks $\{t_l\}_{l=1}^L$. $\mathcal{T}$ extracts high-level features from the content image, aligns them to the style embedding $\boldsymbol{z}_s$ and maps the resulting features to the pixel domain. The style of $\boldsymbol{i}_s$ encapsulated in $\boldsymbol{z}_s$ is transferred to the content image using CIN. This is applied to a particular set of layers to impose the target texture and color scheme by aligning feature maps at different scales.

We define the set of layers where CIN is applied as $\mathcal{A}$. The normalization imposed via CIN consists of an extended form of instance normalization where the target mean and standard deviation are extracted from a style representation $\boldsymbol{z}$.

Given an input $\boldsymbol{i} \in \mathbb{R}^{C \times H \times W}$ and a style representation $\boldsymbol{z} \in \mathbb{R}^D$, CIN is defined as

$$\hat{\boldsymbol{i}} = \text{CIN}(\boldsymbol{i}, \boldsymbol{z}) \in \mathbb{R}^{C \times H \times W} \tag{5}$$

$$\hat{\boldsymbol{i}}^{(k)} = \gamma^{(k)}(\boldsymbol{z})\left(\frac{\boldsymbol{i}^{(k)} - \mathbb{E}[\boldsymbol{i}^{(k)}]}{\sigma(\boldsymbol{i}^{(k)})}\right) + \lambda^{(k)}(\boldsymbol{z}) \tag{6}$$

where $\boldsymbol{i}^{(k)}$, $k \in \{1, \ldots, C\}$ corresponds to the $k$-th input channel, and the sample mean $\mathbb{E}[\boldsymbol{i}^{(k)}]$ and standard deviation $\sigma(\boldsymbol{i}^{(k)})$ are computed along its spatial support. Here, $\gamma^{(k)}, \lambda^{(k)} : \mathbb{R}^D \mapsto \mathbb{R}$ are trainable functions that predict scaling and offset values from the latent representation $\boldsymbol{z}$ for the $k$-th input channel. The layers in $\mathcal{T}$ are characterized by

$$\phi_{c,l} = \begin{cases} \text{CIN}_l\big(t_l(\phi_{c,l-1}), \boldsymbol{z}_s\big), & l \in \mathcal{A} \\ t_l(\phi_{c,l-1}), & l \notin \mathcal{A} \end{cases} \tag{7}$$

where the input of the stylization network corresponds to $\phi_{c,0} = \boldsymbol{i}_c$. The subscript $l$ in the CIN operation indicates that each layer has its own $\gamma$ and $\lambda$ functions to normalize features independently.

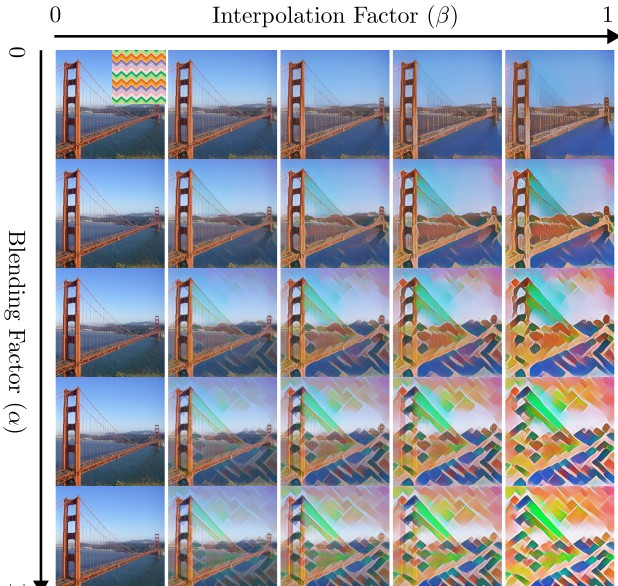

Figure 2: **Feature blending and image interpolation**. A fine-grained control over the final stylized image is obtained via interpolation and blending factors $\alpha$ and $\beta$ that operate in the feature and pixel domains. This prevents augmented views from losing semantic information due to strong transformations.

Our method uses the Fast Style Transfer algorithm. Given its generalization properties and low-dimensional style representations, it is a good match for our framework, where style representations from multiple domains must be efficiently extracted, manipulated and transferred.

## 4 Proposed Method: SASSL

We provide a detailed description of our SSL augmentation technique. First, we break down SASSL's key components and hyperparameters. Then, we tackle the problem of making the augmented images more diverse by utilizing style references from different domains in an efficient manner.

**Algorithm 1:** Style transfer augmentation block

**Input:** $\boldsymbol{i}_c, \boldsymbol{i}_s, \mathcal{F}, \mathcal{T}, \alpha_{\min}, \alpha_{\max}, \beta_{\min}, \beta_{\max}$
**Output:** $\boldsymbol{i}_{\mathrm{cs}}$

$\boldsymbol{z}_c \leftarrow \mathcal{F}(\boldsymbol{i}_c)$ ; # *Style representation of content image*
$\boldsymbol{z}_s \leftarrow \mathcal{F}(\boldsymbol{i}_s)$ ;    # *Style representation of style image*

$\alpha \sim \mathcal{U}(\alpha_{\min}, \alpha_{\max})$ ;              # *Blending factor*
$\hat{\boldsymbol{z}} \leftarrow (1-\alpha)\boldsymbol{z}_c + \alpha\boldsymbol{z}_s$ ;           # *Feature blending*

$\hat{\boldsymbol{i}}_{\mathrm{cs}} \leftarrow \mathcal{T}(\boldsymbol{i}_c, \hat{\boldsymbol{z}})$;
$\beta \sim \mathcal{U}(\beta_{\min}, \beta_{\max})$ ;              # *Interpolation factor*
$\boldsymbol{i}_{\mathrm{cs}} \leftarrow (1-\beta)\boldsymbol{i}_c + \beta\hat{\boldsymbol{i}}_{\mathrm{cs}}$ ;             # *Stylized image*

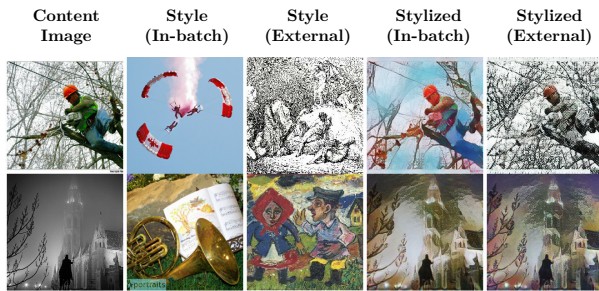

Figure 3: **Style Transfer examples**. Views generated using style references from the same domain *(in-batch)* as well as other domains *(external)*. Stylization obtained using a blending factor $\alpha = 0.5$.

**Style Transfer as data preprocessing.** We incorporate Style Transfer to the default preprocessing pipeline of SSL methods. It is worth noting that SASSL is not specific to a particular SSL approach, and can be readily applied with different methods. Figure 1 shows an example of our augmentation pipeline, where Style Transfer is applied after random cropping. A raw input image $\boldsymbol{i}_0$ is cropped, producing a view that is taken as the content image $\boldsymbol{i}_c$. Given an arbitrary style image $\boldsymbol{i}_s$ (we discuss the choice of $\boldsymbol{i}_s$ below), the Style Transfer block generates a stylized image $\boldsymbol{i}_{\mathrm{cs}}$ by imposing the texture of $\boldsymbol{i}_s$ over $\boldsymbol{i}_c$. Finally, the stylized image $\boldsymbol{i}_{\mathrm{cs}}$ is passed to the remaining augmentation blocks to produce an augmented sample $\boldsymbol{i}_{\mathrm{aug}}$.

As discussed in recent work on SSL augmentation (Han et al., 2022; Chen et al., 2020a), adding a strong transformation to a self-supervised method tends to degrade performance. For this reason, it is crucial to control the amount of stylization imposed in the augmentation stage. We do so by introducing three hyperparameters: probability $p \in [0, 1]$, which dictates whether an image is stylized or not, a blending factor $\alpha \in [0, 1]$ to combine content $\boldsymbol{z}_c$ and style $\boldsymbol{z}_s$ representations, and an interpolation factor $\beta \in [0, 1]$ to combine content $\boldsymbol{i}_c$ and stylized $\hat{\boldsymbol{i}}_{\mathrm{cs}}$ images.

Given style representations extracted from content and style images $\boldsymbol{z}_c = \mathcal{F}(\boldsymbol{i}_c)$ and $\boldsymbol{z}_s = \mathcal{F}(\boldsymbol{i}_s)$, respectively, we obtain an intermediate stylized image $\hat{\boldsymbol{i}}_{\mathrm{cs}}$ by applying a convex combination based on blending factor $\alpha$.

$$\hat{\boldsymbol{i}}_{\mathrm{cs}} = \mathcal{T}(\boldsymbol{i}_c, \hat{\boldsymbol{z}}), \quad \hat{\boldsymbol{z}} = (1-\alpha)\boldsymbol{z}_c + \alpha\boldsymbol{z}_s \tag{8}$$

Then, the final stylization output is obtained via a convex combination between the intermediate stylized image $\hat{\boldsymbol{i}}_{cs}$ and the content image $\boldsymbol{i}_c$ based on interpolation factor $\beta$.

$$\boldsymbol{i}_{\mathrm{cs}} = (1-\beta)\boldsymbol{i}_c + \beta\hat{\boldsymbol{i}}_{\mathrm{cs}} \tag{9}$$

Algorithm 1 describes our proposed Style Transfer data augmentation block. Figure 2 illustrates the effect of the feature blending and image interpolation operations, showcasing their importance to control the stylization effect without degrading the semantic attributes.

SASSL operates over minibatches, allowing efficient data pre-processing. Let $\boldsymbol{I}_c \in \mathbb{R}^{B \times C \times H_c \times W_c}$ and $\boldsymbol{I}_s \in \mathbb{R}^{B \times C \times H_s \times W_s}$ be content and style minibatches, respectively, comprised by $B$ images $\boldsymbol{I}_c^{(b)}$ and $\boldsymbol{I}_s^{(b)}, b \in \{1, \ldots, B\}$. Then, the stylized minibatch $\boldsymbol{I}_{\mathrm{cs}} \in \mathbb{R}^{B \times C \times H_c \times W_c}$ is generated by applying Style Transfer between a sample from the content batch and a sample from the style batch, given an arbitrary selection criterion. We propose two alternatives for selecting style images to balance between augmentation diversity and efficiency.

**Diversifying style references.** In contrast to traditional data augmentation, Style Transfer can leverage a second dataset to extract style references. This opens the possibility of selecting style images from different domains, diversifying the transformations applied to the pretraining dataset. SASSL relies on two approaches for sampling style references: *external* and *in-batch* stylization.

*External* stylization consists on pre-computing representations of an arbitrary style dataset and sampling from them during pretraining. This allows controlling the styles to impose on the augmented views while

reducing the computational overhead of Style Transfer. Under this configuration, the Style Transfer block receives a content minibatch along with a minibatch of pre-computed style representations extracted from an arbitrary style dataset, and generates a minibatch of stylized images.

On the other hand, *in-batch* stylization uses the styles depicted in the content dataset itself by using other images of the content minibatch as style references. This is of particular interest for large-scale pretraining datasets covering multiple image categories and thus textures (e.g. ImageNet). So, enabling the use of a single dataset for both pretraining and stylization is a valid alternative.

Following this, samples from the same minibatch can be used as style references by associating pairs of images in a circular fashion. More precisely, a style minibatch $I_s \in \mathbb{R}^{B \times C \times H_c \times W_c}$ is generated by applying a circular shift on the content minibatch indices

$$I_s^{(b)} = I_c^{((b-b_0) \bmod B)} \tag{10}$$

where mod denotes the modulo operation and $b_0$ is an arbitrary offset. Figure 3 shows ImageNet samples stylized using pre-computed style representations from the Painter by Numbers dataset (Kan, 2016) via *external* stylization, as well as using other ImageNet samples taken from the content minibatch via *in-batch* stylization.

## 5 Experiments

### 5.1 Downstream Task Performance

We evaluate the downstream ImageNet classification accuracy of SSL models pretrained via SASSL on the MoCo framework. We compare a MoCo v2 model pretrained with our data augmentation vs. a MoCo v2 baseline with default augmentation (Chen et al., 2020b). Note that MoCo v2 and SimCLR use the same loss, architecture, and augmentations (they differ by MoCo's momentum encoding).

Table 1: **SASSL + MoCo v2 downstream classification performance on ImageNet**. Linear probing accuracy (%) of a ResNet-50 backbone pretrained using SASSL + MoCo v2. Mean accuracy reported over five random trials.

| Method | Top-1 Acc. | Top-5 Acc. |
|---|---|---|
| MoCo v2 (Default) | 72.55 | 91.19 |
| SASSL + MoCo v2 **(Ours)** | **74.64** | **91.68** |

**Pretraining settings.** Our pretraining setup is similar to the canonical SSL setup used to pretrain SimCLR and BYOL. We use the same loss, architecture, optimizer, and learning rate schedule as MoCo v2 for fair comparison. We pretrain a ResNet-50 encoder on ImageNet for $1,000$ epochs via SASSL. To measure downstream accuracy, we add a linear classification head on top of the pretrained backbone and train in a supervised fashion on ImageNet.

SASSL pretraining applies Style Transfer only to the left view (no changes in augmentation are applied to the right view). It is applied with a probability $p = 0.8$ using blending and interpolation factors drawn from a uniform distribution $\alpha, \beta \sim \mathcal{U}(0.1, 0.3)$. We found that this modest stylization best complimented the existing augmentations, avoiding overly-strong transformations that can hinder performance (Han et al., 2022; Chen et al., 2020a).

**Results.** Table 1 compares the downstream classification accuracy obtained by our SASSL augmentation approach on MoCo v2 using *external* stylization from the Painter by Numbers dataset. Results indicate our proposed augmentation improves downstream task performance by 2.09% top-1 accuracy. This highlights the value of Style Transfer augmentation in self-supervised training, where downstream task performance significantly boosts by incorporating transformations that decouple content and style. We also report results with *in-batch* stylization in Section 5.5.

### 5.2 Transfer Learning Performance

To better understand the robustness and generalization of image representations learned using our proposed data augmentation approach, we evaluate its transfer learning performance across various tasks. By incorporating Style Transfer, we hypothesize that the learned representations become invariant to changes

Table 2: **Transfer learning performance.** Downstream top-1 classification accuracy (%) of SASSL + MoCo v2 pretrained on ImageNet. Our data augmentation method generates specialized image representations that improve transfer learning performance, as shown in linear probing and fine-tuning scenarios. Mean accuracy reported over five random trials.

| | | ImageNet | ImageNet-1% | iNat21 | Retinopathy | DTD | Food101 | CIFAR10 | CIFAR100 | SUN397 | Cars | Caltech-101 | Flowers |
|---|---|---|---|---|---|---|---|---|---|---|---|---|---|
| | | | | | | | *Linear Probing* | | | | | | |
| | None (Default) | 72.55 | 53.23 | 41.33 | **75.88** | 72.68 | 73.82 | 89.94 | 71.93 | 69.96 | 53.15 | 88.19 | 93.39 |
| | ImageNet (**Ours**) | 74.07 | 56.87 | 45.01 | 75.75 | 73.69 | 74.43 | 90.93 | 73.26 | 69.67 | **64.87** | 89.3 | 95.27 |
| | iNat21 (**Ours**) | 74.28 | 56.76 | 44.70 | 75.75 | 72.75 | 74.3 | **91.04** | 73.29 | **70.07** | 63.96 | **89.89** | 94.7 |
| | Retinopathy (**Ours**) | 74.02 | **56.99** | 44.9 | 75.78 | 73.73 | 74.53 | 90.8 | 73.3 | 69.63 | 64.06 | 89.17 | 94.94 |
| | DTD (**Ours**) | 74.32 | 56.77 | **45.08** | 75.76 | **74.41** | **74.88** | **91.04** | **73.41** | 69.71 | 64.58 | 89.3 | 95.24 |
| | PBN (**Ours**) | **74.64** | 56.9 | 45.02 | 75.79 | 72.77 | 74.37 | 90.85 | 73.38 | 69.69 | 64.12 | 89.59 | **95.45** |
| | | | | | | | *Fine-tuning* | | | | | | |
| | None (Default) | 74.89 | 51.61 | 77.92 | 78.89 | 71.54 | 87.25 | 96.91 | 83.4 | 74.25 | 83.63 | 89.27 | 95.75 |
| | ImageNet (**Ours**) | 75.52 | 51.74 | 79.21 | 79.64 | **72.31** | 87.48 | 97.0 | 83.21 | 73.89 | **90.33** | 88.26 | **96.6** |
| | iNat21 (**Ours**) | **75.58** | 51.86 | 79.19 | 79.6 | 71.35 | 87.4 | **97.05** | 83.29 | 74.05 | 90.04 | 88.55 | 95.76 |
| | Retinopathy (**Ours**) | 75.52 | 51.76 | 79.23 | 79.63 | 72.07 | 87.39 | 96.97 | **83.68** | **74.26** | 89.96 | 88.44 | 96.34 |
| | DTD (**Ours**) | 75.24 | 51.73 | **79.24** | **79.7** | 70.59 | **87.66** | 96.77 | 83.28 | 74.17 | 89.59 | **89.54** | 95.59 |
| | PBN (**Ours**) | 75.05 | 51.85 | 79.2 | 79.63 | 71.35 | 87.56 | 96.97 | 83.36 | 74.18 | 89.75 | 88.97 | 95.77 |

*(Row group label at left: **Style Dataset**; column header spanning the dataset columns: **Target Dataset**)*

in appearance such as color and texture. This forces the feature extraction process to rely exclusively on semantic attributes. As a result, the learned representations may become more robust to domain shifts, improving downstream task performance across datasets. We empirically show this property by evaluating the transfer learning performance of image representations trained using SASSL under linear probing and fine-tuning scenarios.

**Downstream settings.** We compare the transfer learning accuracy of ResNet-50 pretrained via MoCo v2 using SASSL against a MoCo v2 baseline with default data augmentation. The evaluated models are pretrained on ImageNet and transferred to eleven target datasets: ImageNet-1% subset (Chen et al., 2020a), iNaturalist '21 (iNat21) (iNaturalist 2021), Diabetic Retinopathy Detection (Retinopathy) (Kaggle & Eye-Pacs, 2015), Describable Textures Dataset (DTD) (Cimpoi et al., 2014), Food101 (Bossard et al., 2014), CIFAR10/100 (Krizhevsky, 2009), SUN397 (Xiao et al., 2010), Cars (Krause et al., 2013), Caltech-101 (Fei-Fei et al., 2004), and Flowers (Nilsback & Zisserman, 2008).

To have a clear idea of the effect of the style dataset in SASSL's pipeline, we pretrain five ResNet-50 backbones, each using a different style. We use ImageNet, iNat21, Retinopathy, DTD, and Painter by Numbers (PBN) as style datasets. More precisely, we transfer five models, each pretrained on a different style, to each of eleven target datasets. We also include ImageNet as target dataset to compare the effect of different styles on downstream task performance. This leads to 60 transfer learning scenarios used to better understand the effect of various styles on different image domains.

Transfer learning is evaluated in terms of top-1 classification accuracy on linear probing and fine-tuning. All models were pretrained as described in Section 5.1. We report mean accuracy across five trials. Please refer to Appendix A.4 for full linear probing and fine-tuning training and testing settings.

**Results.** Table 2 shows the top-1 classification accuracy obtained via transfer learning. For linear probing, SASSL significantly improves the average performance on eleven out of twelve target datasets by up to 10% top-1 classification accuracy. For Retinopathy, SASSL obtains on-par linear probing accuracy to the default MoCo v2 model.

For fine-tuning, all models trained via SASSL outperform the baseline. Results show the average top-1 classification accuracy improves by up to 6%. This suggests SASSL generalizes across datasets, spanning from textures (DTD) to medical images (Retinopathy). Note that, for a fair comparison, we do not perform hyperparameter tuning. Interestingly, the relative performance obtained from different style datasets generally differs comparably to the measurement uncertainty, which is shown for these experiments in Section A.5 of the supplementary material. This suggests that the choice of style dataset is secondary in importance, while the main benefit comes from the use of SASSL itself.

## 5.3 Few-shot Learning Performance

To further demonstrate the representation learning capabilities of the data augmentation imposed via SASSL, we conduct experiments on *few-shot classification*. We compare our ResNet-50 backbone pretrained via SASSL + MoCo v2 against a MoCo v2 baseline in the context of one and ten-shot learning on ImageNet.

Table 3: **SASSL + MoCo v2 Few-shot learning performance**. One and ten-shot top-1 classification accuracy (%) of representations learned via SASSL + MoCo v2. Accuracy reported on a single trial.

| Method | One-shot Acc. | Ten-shot Acc. |
|---|---|---|
| MoCo v2 (Default) | 19.56 | 45.05 |
| SASSL + MoCo v2 (**Ours**) | **20.55** | **46.73** |

Table 3 shows the few-shot classification accuracy. Results reveal that SASSL boosts few-shot classification top-1 accuracy by over 1% in both one and ten-shot learning. This aligns with our previous experiments, suggesting that SASSL promotes more general image representations.

## 5.4 Additional Downstream Performance Evaluation

**Performance on other SSL methods.** To assess SASSL's broader impact, we evaluate its effectiveness on two other SSL methods, SimCLR and BYOL. We pretrain ResNet-50 backbones with each method, and then use linear probing on ImageNet to compare the quality of their learned representations. For each method, default pretraining and linear probing configurations are used. For SASSL, we employ its recommended hyperparameters ($\alpha, \beta \in [0.1, 0.3]$, $p = 0.8$) and PBN as style dataset.

Table 4 shows the accuracy attained by SimCLR and BYOL equipped with SASSL. Results show our proposed data augmentation technique boosts top-1 accuracy by approximately 1% in both cases, highlighting its potential across multiple SSL techniques.

Table 4: **SASSL downstream performance using alternative SSL methods**. Linear probing accuracy (%) on ImageNet using a ResNet-50 backbone pretrained via SimCLR and BYOL. Accuracy reported on a single trial.

| Method | Top-1 Acc. | Top-5 Acc. |
|---|---|---|
| SimCLR (Default) | 68.62 | 88.7 |
| SASSL + SimCLR (**Ours**) | **69.58** | **89.01** |
| BYOL (Default) | 74.09 | 91.83 |
| SASSL + BYOL (**Ours**) | **75.13** | **92.12** |
| SwAV (Default) | 70.45 | 89.6 |
| SASSL + SwAV (**Ours**) | **71.3** | **90.39** |

Table 5: **SASSL + MoCo downstream performance using alternative backbones**. Linear probing accuracy (%) on ImageNet using ResNet-50 (x4) and ViT-B/16 representation models. Accuracy reported on a single trial.

| Backbone | Method | Top-1 Acc. | Top-5 Acc. |
|---|---|---|---|
| ResNet-50 x4 (375M) | MoCo v2 (Default) | 77.2 | 93.32 |
| | SASSL + MoCo v2 (**Ours**) | **78.21** | **93.98** |
| ViT-B/16 (86M) | MoCo v3 (Default) | 75.01 | 92.43 |
| | SASSL + MoCo v3 (**Ours**) | **75.51** | **92.56** |

**Performance on other representation models.** We explore SASSL's performance on models with varying complexity and architecture. For complexity, we employ ResNet-50 (x4), a scaled-up version of the previously evaluated ResNet-50 (from 24 to 375 million parameters). This allows us to probe how SASSL scales with increased model size. In terms of architecture, we employ ViT-B/16, a Transformer-based backbone with 86 million parameters and a distinct design compared to previous CNN models.

We pretrain and linearly probe a ResNet-50 (x4) representation model on ImageNet via MoCo v2. Pretraining and downstream settings follow our default configuration, as documented in the Appendices A.3 and A.4. Similarly, we pretrain and linear probe a ViT-B/16 model on ImageNet via MoCo v3. In this case, SASSL employed a blending factor $\alpha$ uniformly sampled between 0.1 and 0.5.

Table 5 reports the downstream classification accuracy for ResNet-50 (x4) and ViT-B/16. ResNet-50 (x4) results show SASSL improves top-1 classification accuracy by 1.1%, mirroring its earlier improvement. Similarly, ViT-B/16 results show SASSL improves top-1 accuracy by 0.5%. These suggest that SASSL is not limited to CNN backbones, but can also be extended to ViTs. While this margin is currently smaller for ViTs, we emphasize that no hyperparameter tuning was employed in these experiments.

## 5.5 Ablation Studies

To shed light on how SASSL affects accuracy on ImageNet, we break down its components and assess individual contributions to downstream performance. We also study how aligning different layers in the stylization network $\mathcal{T}$ boosts accuracy. See Appendices A.6 and A.7 for additional ablations and SASSL's computational requirements.

**SASSL components.** For the ablation study, we cover four cases: (i) MoCo v2 with default augmentation, (ii) SASSL + MoCo v2 using in-batch representation blending and no pixel interpolation ($\beta = 1$), (iii) SASSL + MoCo v2 using in-batch representation blending and pixel interpolation, and (iv) SASSL + MoCo v2 using all its attributes (blending, interpolation and an external style dataset).

Table 6 shows our ablation study using MoCo v2 as SSL technique. Results highlight the importance of controlling the amount of stylization using both representation blending and image interpolation. Without image interpolation, using Style Transfer as data augmentation degrades the downstream classification performance by more than 1.5% top-1 accuracy.

On the other hand, by balancing the amount of stylization via blending and interpolation, SASSL boosts performance by more than 1.5%. This is a significant improvement for the challenging ImageNet scenario. Finally, by incorporating an external style dataset such as PBN, we further improve downstream task performance by almost 2.1% top-1 accuracy. This shows the importance of diverse style references and their effect on downstream tasks.

Table 6: **Ablation study.** Linear probing accuracy (%) for representations learned via SASSL under different configurations. Mean accuracy reported over five random trials.

| Method | Configuration | Style | Top-1 Acc. | Top-5 Acc. |
|---|---|---|---|---|
| MoCo v2 (Default) | – | – | 72.55 | 91.19 |
| SASSL + MoCo v2 (**Ours**) | $p = 0.8,$ $\alpha \in [0.1, 0.3]$ $\beta = 1$ | ImageNet (*in-batch*) | 70.87 | 89.33 |
| | $p = 0.8,$ $\alpha \in [0.1, 0.3]$ $\beta \in [0.1, 0.3]$ | ImageNet (*in-batch*) | 74.07 | 91.58 |
| | $p = 0.8,$ $\alpha \in [0.1, 0.3]$ $\beta \in [0.1, 0.3]$ | PBN (*external*) | **74.64** | **91.68** |

Table 7: **Effect of the number of stylized layers in downstream performance.** Linear probing classification accuracy (%) of a ResNet-50 model pretrained via SASSL + MoCo v2, where Style Transfer is applied using a subset of the available layers. Accuracy reported on a single trial.

| Method | Stylized Layers | Top-1 Acc. | Top-5 Acc. |
|---|---|---|---|
| MoCo v2 (Default) | – | 72.97 | 90.86 |
| SASSL + MoCo v2 (**Ours**) | None ($\hat{z} = z_c$) | 73.77 | 91.64 |
| | First 4 layers | 73.75 | 91.58 |
| | First 8 layers | 74.09 | 91.76 |
| | First 10 layers | 74.27 | 91.74 |
| | All (13 layers) | **75.38** | **92.21** |

**Number of stylized layers.** We explore how the number of layers used to apply style transfer via CIN affects downstream performance. We analyze three cases: (i) stylizing using the first two residual blocks of the Stylization Network $\mathcal{T}$ (four layers from blocks 1 and 2), (ii) the first four residual blocks (eight layers from blocks 1 to 4), and (iii) all five residual blocks (ten layers).

For each case, we pretrain and linearly probe a ResNet-50 on ImageNet using SASSL + MoCo v2 with its recommended settings ($\alpha, \beta \in [0.1, 0.3]$, $p = 0.8$) and PBN as style dataset. To fully remove the effect of a style embedding $z_s$, our comparison includes a model pretrained using the content image itself as style reference ($\hat{z} = z_c$). We also compare our full SASSL + MoCo v2 model, stylizing all residual and upsampling blocks of $\mathcal{T}$ (thirteen layers).

Table 7 shows a progressive enhancement in accuracy with increasing stylization depth. Adding stylization to the first four layers showed negligible gains, mirroring the accuracy of the unaligned model. Stylizing the first eight and ten layers yielded modest improvements of 0.34% and 0.52%, respectively, implying a growing influence of deeper layers on accuracy. Notably, pretraining with full stylization, encompassing both residual and upsampling layers, attains a 1.61% accuracy boost, suggesting the importance of aligning deeper upsampling layers for downstream performance.

# 6 Conclusion

We propose SASSL, a novel data augmentation approach based on Neural Style Transfer that exclusively transforms the style of training samples, diversifying data augmentation during pretraining while preserving semantic attributes. We empirically show our approach outperforms well-established methods such as MoCo v2, SimCLR and BYOL by up to 2% top-1 classification accuracy on ImageNet. SASSL also improves the transfer capabilities of learned representations, enhancing linear probing and fine-tuning performance across domains by up to 10% and 6% top-1 accuracy, respectively. Our technique can be extended to other SSL methods and models with minimum hyperparameter changes, as experimentally shown.

## Broader Impact Statement

This work proposes a novel data augmentation approach leveraging Neural Style Transfer to enhance Self-supervised Learning, particularly for domains with limited data or expensive annotations. Our method utilizes semantic-aware image preprocessing to extract robust representations that generalize across diverse domains. This advancement tackles the critical challenge of using unlabeled data for Deep Learning, which has many potential positive impacts in both technical and societal fronts. SASSL's style transfer component helps to reduce sensitivity to image texture, potentially improving model robustness to texture bias. However, since our method primarily modifies data augmentation, it may not fully address other potential biases arising from pre-training datasets or learning strategies.

## Acknowledgements

The authors would like to thank Arash Afkanpour and Luyang Liu for their insightful comments and feedback.

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
