# OpenReview forum: "SASSL: Enhancing Self-Supervised Learning via Neural Style Transfer"
_TMLR — Accepted by TMLR_

### Review · Reviewer_pV6L · 2024-06-21

**Summary Of Contributions:**

The authors investigate the impacts of using stylization in the data augmentation pipeline of contrastive self-supervised learning frameworks like SimCLR, BYOL, and MoCo v2 and report gains over the baseline frameworks.

**Audience:**

Yes

**Broader Impact Concerns:**

Filtering out the style datasets so that they don't include inappropriate or licensed images could be important. The latter could lead to concerns about copyright infringement.

**Claims And Evidence:**

Yes

**Requested Changes:**

* In the abstract, it would be better to give the readers a hint about the changes in training throughput because of stylization (if there are any). My point about including some information on the timing-related ramifications stands for other areas of the paper, such as the introduction, results, etc. Additionally, directly mentioning the gains obtained in the downstream performance in the abstract would be informative.

* Section 2.1 misses the multi-crop augmentation policy introduced in SwAV [1] and is being used in DINO [2] and other works like [3], [4].

* The preliminaries on SSL could be categorized into contrastive and non-contrastive approaches rather than being restricted to SimCLR only. This gives readers a better view of the field.

* Could include a section studying the implications of stylization in the context of non-contrastive frameworks.

* Some more ablations could be:
  * Does increasing the probability of stylization during the latter half of the training help improve the performance? It's inspired by the EMA decaying scheme, which is usually followed in works like data2vec [5].
  * Why do we need a circular scheme when using the in-batch stylization?
  * Can we use a blend of in-batch and external dataset stylization?
  * Can we use an aggregate dataset sourced by combining images from all the external datasets used for stylization?
  * Why is only MoCo v2 used for reporting downstream performance?

* Details that could be added:
  * It Could be worth detailing what the "1%" metric means in Table 2.
  * Highlight the importance of using different stylization datasets for different downstream datasets. Table 2 confirms this phenomenon.
  * The ablation in "Number of stylized layers" is not fully clear to me. Does it mean fine-tuning the stylization network along with the base model being pre-trained?

## References

[1] Unsupervised Learning of Visual Features by Contrasting Cluster Assignments, https://arxiv.org/abs/2006.09882.

[2] Emerging Properties in Self-Supervised Vision Transformers, https://arxiv.org/abs/2104.14294.

[3] Self-supervised Pretraining of Visual Features in the Wild, https://arxiv.org/abs/2103.01988.

[4] Vision Models Are More Robust And Fair When Pretrained On Uncurated Images Without Supervision, https://arxiv.org/abs/2103.01988.

[5] data2vec: A General Framework for Self-supervised Learning in Speech, Vision and Language, https://arxiv.org/abs/2202.03555.

**Strengths And Weaknesses:**

## Strengths

* The paper is well-written and easy to follow.
* The results have been presented in a clear format so that the reader can understand the gains.

## Weaknesses

* Effects of the proposed method on non-contrastive self-supervised methods like SwAV [1] are missing.
* Ablations on the important degrees of freedom introduced in the proposed method are missing.
* Training throughput information is missing. I think this is important to mention because stylization can incur an additional increase in the preprocessing time.

## References

[1] Unsupervised Learning of Visual Features by Contrasting Cluster Assignments, https://arxiv.org/abs/2006.09882.

---

> ### Author Response · Authors · 2024-10-03
> **Response to reviewer pV6L**
>
> We thank the reviewer for their valuable feedback.  In our response, we detail the steps we have taken to address the current concerns held by the reviewer.
>
> **SwAV:** We have added experimental results with SwAV to Table 4, supplementing the non-contrastive SSL methods like BYOL that were there previously.
>
> **Training Throughput:** Because style augmentation can be performed asynchronously to training (like other augmentations), and because it does not alter the input shape of the resulting images, it does not affect the throughput during pretraining. We are grateful to the reviewer for calling our attention to this, as we think this is a significant advantage of this approach compared to alternatives which may require further compute in the model. We have added a line to the end of the abstract to highlight this feature.
>
> **Reference to Multicrop:** We have added the suggested reference to section 2.1
>
> **Discussion of broader SSL in preliminaries:** Our main purpose with using SimCLR as the illustrative example is that the template originally proposed in that work has largely persisted across SSL, even to this day. We do agree however that we should not emphasize the contrastive nature of SimCLR, as follow up works have shown that this is not a requirement for good performance, and have modified the manuscript to focus less on this particular property (though we still use SimCLR as an example to introduce the SSL template that we follow).
>
> **Implication of Stylization in non-contrastive frameworks:** We did not find a significant difference between the impact of stylization in contrastive vs non-contrastive frameworks, and therefore do not think that this warranted special discussion.
>
> **Additional Ablations:**  Regarding the additional exploration regarding stylization probability, blending, and dataset combinations, we note that these suggestions are not strictly speaking ablations, but rather further extensions to this work. While they can certainly lead to performance improvement over the current results, they would require additional engineering and compute to facilitate, and therefore we found them out of scope of this study and decided to prioritize other experiments like the addition of evaluations using SwAV.
>
> Our ablations show that we do not need a circular scheme, as randomly selecting the style image from the batch achieved performance consistent with the reported numbers to within our margin of error. However, there is still appeal to this scheme in practice: it is equivalent to randomly selecting the style image from the batch (due to the shuffling of the dataset), but guarantees that the content image is not the style image.  Our existing experiments had found that using the input image as its own style image achieved lower performance, indicating that it is safer to use a technique like the circular scheme because it guarantees that each image sees a different image as its style reference.
>
> We reported the downstream performance for other SSL methods like BYOL and SimCLR in Table 4, and have added SwAV. Due to space and resource constraints, it was not feasible for us to provide the same ablations and detailed analysis for all methods, and we opted to focus on MoCo v2.
>
> **Additional Details:** We thank the reviewer for bringing these to our attention and have added all of the requested details. To summarize here:
>
> >It Could be worth detailing what the "1%" metric means in Table 2
>
> We apologize for the confusion. ImageNet 1% refers to a few-shot ImageNet containing only 1% of the dataset.  For this, we used the implementation from `tensorflow_datasets`.  We make a note of this in the main text, but to add further clarity to the Table, we have changed the name to ImageNet-1% rather than ImageNet (1%) to indicate that the 1% is part of the name of the dataset.
>
> > Highlight the importance of using different stylization datasets for different downstream datasets.
>
> As we noted above, different style datasets perform equivalently to within our margin of error.  We had not noted this in the manuscript and acknowledge that this was not clear from looking at Table 2 since we could not fit our measurement uncertainties in that table.  With the text that we have added to the manuscript, we have made sure to mention this in the interest of clarity.
>
> >The ablation in "Number of stylized layers" is not fully clear...
>
> The “number of stylized layers” ablation studied the impact of applying Conditional Instance Normalization using only a subset of the available layers where it is typically applied in the stylization network.  We agree that this is not clear from the description in Table 7, and have clarified this in the caption.  The main text appears to also have some ambiguous wording, and we have made a point to correct this as well.
>
> Once again, we are grateful to the reviewer for their feedback, and we look forward to continuing the discussion and answering any follow up questions.

---

### Review · Reviewer_hgHu · 2024-07-04

**Summary Of Contributions:**

This paper proposes SASSL (Style Augmentations for Self-Supervised Learning), a novel data augmentation technique for self-supervised learning (SSL) that leverages Neural Style Transfer to generate semantically consistent augmented samples. SASSL decouples semantic and stylistic attributes in images and applies transformations exclusively to their style while preserving content, resulting in diverse samples that better retain semantic information. The authors show that SASSL boosts downstream classification performance on ImageNet by up to 2% compared to established SSL methods like MoCo, SimCLR, and BYOL, while achieving superior transfer learning results across various datasets.

**Audience:**

Yes

**Broader Impact Concerns:**

The paper includes a Broader Impact Statement. The statement could include a description of potential bias learned by the model, especially in self-supervised models, where the bias could be learned implicitly.

**Claims And Evidence:**

Yes

**Requested Changes:**

My main concern is related to the choice of the stylization dataset, where a more thorough discussion is needed in my opinion. It does not hinder the proposed method's strengths but it limits its applicability.

**Strengths And Weaknesses:**

## Strengths
1. **Improved downstream performance**: SASSL boosts top-1 image classification accuracy on ImageNet by up to 2% compared to established SSL methods like MoCo, SimCLR, and BYOL. It also achieves superior transfer learning performance across various datasets.
2. **Generalization across datasets**: Results show SASSL generalizes across datasets, spanning from textures (DTD) to medical images (Retinopathy), with linear probing accuracy improving by up to 10% and fine-tuning accuracy by up to 6%.
3. **Effectiveness on different SSL methods**: SASSL boosts top-1 accuracy by approximately 1% when applied to SimCLR and BYOL, showing its potential across multiple SSL techniques.
4. **Improved few-shot learning**: SASSL learns more general image representations, boosting few-shot classification top-1 accuracy by over 1% in both one and ten-shot learning on ImageNet.
5. **Flexibility with representation models**: SASSL achieves consistent improvements when applied to different models and architectures, such as ResNet-50 and ViT-B/16.

## Weaknesses
1. **Selection of stylization:** Although the authors provide in the supplementary a way to determine the similarity between the datasets, it is not stated or presented a way to select the optimal dataset to use as style. it seems one needs to try more than one and look at the test accuracy, defeating the purpose of the test set. according to the t-sne plot in the supplementary, it seems that, apart from the Diabetic Retinopathy dataset, the rest of the datasets are very close in this embedding space, making the choice even more difficult. a more thorough discussion of the selection process would be beneficial.
2. **Image-to-image translation:** Although the proposed SASSL reaches reasonable results on the proposed benchmark, the paper lacks a discussion of more recent approaches to image stylization/transformation, such as image-to-image generation, which diffusion models are good at. One notable example is InstructPix2Pix [1], where the image can be stylized or transformed by describing via text the desired transformation. While I am not asking to run this pipeline, I would like a discussion of the advantages/disadvantages of the proposed method w.r.t. these approaches.

---

> ### Author Response · Authors · 2024-10-03
>
> We are grateful to the reviewer for taking the time to review our work and for providing their feedback.  We are happy to address the remaining concerns held by the reviewer, and will respond to each point individually to hopefully ameliorate these concerns.
>
> **Style dataset selection:** We agree that a broader discussion of style dataset selection is interesting, though we regret that we are unable to provide meaningful recommendations based on the results of our experiments.  In particular, we found that all choices of style dataset performed equivalently to within our margin of error (as reported in Tables 10 and 11 of the supplementary material). The main difference was between methods which used SASSL and methods which did not. We agree that this point was underemphasized in the manuscript and have added the following text to the end of Section 5.2 to clarify this point:
>
> “Interestingly, the relative performance obtained from different style datasets generally differs comparably to the measurement uncertainty, which is shown for these experiments in Section A.5 of the supplementary material.  This suggests that the choice of style dataset is secondary in importance, while the main benefit comes from the use of SASSL itself.”
>
> **Image-to-image translation:** We note that the focus of this paper is on using Style transfer as a data augmentation for SSL, and that the discussion of new approaches to style transfer is largely unrelated to the work pursued in this paper.  We can however provide a discussion here on the relative advantages and disadvantages of traditional style transfer methods compared to these newer approaches. First and foremost, methods like InstructPix2Pix leverage label information in the form of captions to determine what aspects are content versus what aspects are style. This differs from the approach we use, which leverages style transfer models that do this in an unlabeled manner. The obvious advantage of the diffusion model is that one can induce particular stylistic changes based on text instructions, while the advantage of the traditional style transfer models is that they do not need to leverage labels and large scale pretraining. Second, InstructPix2Pix, or similar approaches leveraging diffusion, must perform expensive and time consuming generations in order to perform each training step (they report 9 seconds per image in Appendix A.3 in their paper), prohibitively increasing the resource expenditure and decreasing the throughput during pretraining.  In contrast, traditional style transfer methods utilize only a single forward pass of the model, can easily be run asynchronously with insignificant additional compute, and cause no decrease in pretraining throughput.
>
> **Broader Impact Statement:**  We appreciate the reviewer's comments on our broader impact statement. Since SASSL enhances SSL methods by exclusively modifying the data augmentation pipeline, it does not alter the underlying model architecture or pre-training dataset. Therefore, the resulting models may still be susceptible to biases inherent in these components [1, 2, 3].
>
> However, by incorporating style transfer, SASSL potentially mitigates texture bias often associated with large-scale datasets like ImageNet. As suggested by the reviewer, we have added the following explanation to our broader impact statement:
>
> "SASSL's style transfer component helps to reduce sensitivity to image texture, potentially improving model robustness to texture bias. However, since our method primarily modifies data augmentation, it may not fully address other potential biases arising from pre-training datasets or learning strategies."
>
> [1] Wang, Angelina, and Olga Russakovsky. "Overwriting pretrained bias with finetuning data." Proceedings of the IEEE/CVF International Conference on Computer Vision. 2023.
>
> [2] Sirotkin, Kirill, Pablo Carballeira, and Marcos Escudero-Viñolo. "A study on the distribution of social biases in self-supervised learning visual models." Proceedings of the IEEE/CVF Conference on Computer Vision and Pattern Recognition. 2022.
>
> [3] Wang, Xudong, et al. "Debiased learning from naturally imbalanced pseudo-labels." Proceedings of the IEEE/CVF Conference on Computer Vision and Pattern Recognition. 2022.

---

### Review · Reviewer_w3Rd · 2024-09-03

**Summary Of Contributions:**

This work introduces a novel data augmentation method for self-supervised learning using neural style transfer. Most methods for self-supervised learning augment data by incorporating simple transformations such as random cropping, flips, noise addition, and more. Such transformations could destroy semantic information in the image, leading to potential degradation in downstream performance. This work considers using neural style transfer as a technique to augment data in such a way that semantic information is not lost while also allowing for effective data augmentation. Neural style transfer morphs a content image to a particular style using the features of a deep neural network. For faster generation of augmented images, the authors consider autoencoder-based approaches to generate augmented images, as opposed to optimization-based approaches to generate such images. Once generated, these images can then be used in any self-supervised learning training pipeline. The method is shown to improve performance over standard self-supervised methods in image classification along with aiding in transfer learning.

**Audience:**

Yes

**Claims And Evidence:**

Yes

**Requested Changes:**

I believe addressing the following would aid in strengthening the work in my view:

**Main change**
- Could the authors move the discussion of the choice of style for improved performance in the main body?

**Minor questions**
- Do the authors have intuition as to why in certain cases, adding style from the target dataset does not achieve high performance? For example, in Table 2, on ImageNet, other using other datasets for SASSL improved performance over ImageNet.
- I appreciate the ablation study on the interpolation parameter, which matches my intuition that adding too much style can degrade performance. Is there a similar effect on the blending parameter?

**Strengths And Weaknesses:**

**Strengths:**
- The paper is well-written and easy to follow, with all of the details presented in an easy-to-read way.
- The idea of using neural style transfer as an additional data augmentation technique is simple and compelling. Another compelling feature is that NST as data augmentation can effectively be an additional add-on to any standard self-supervised learning method.
- Across many settings, there are clear performance gains.

**Weaknesses:**
- There are a number of hyperparameters one could choose in this method, making it potentially difficult to use right out of the box. The main body of the paper could be improved with a deeper discussion of these choices. For example, a discussion on the choice of style for a given task and dataset is presented in the appendix, but I feel as though it would be more beneficial to have this in the main text. Please see the requested changes section for more.

---

> ### Author Response · Authors · 2024-10-03
> **Response to reviewer w3Rd**
>
> We thank the reviewer for a thoughtful and thorough review.  We agree that the discussion of the choice of style is interesting, however we found in our experiments that the choice of style was largely inconsequential.  Our margin of uncertainty (as shown in Tables 10 and 11) is enough to explain the differences in performance between different choices of style datasets on most tasks.  The main conclusion we can draw is that the use of style transfer itself is impactful, as most runs exhibit statistically significant improvements over the baseline.  The only exception is Linear Probing on Retinopathy, where all methods appear to perform equally.  We attribute this to the significant misalignment between the pretraining dataset (ImageNet) and the downstream dataset (Retinopathy), which we show in Appendix A.2, suggesting that the representations learned via ImageNet pretraining are of limited utility in this case.  Our experiments show that style transfer induces statistically significant improvement for fine-tuned models in that case, adding circumstantial, but not comprehensive empirical evidence for this claim.  Following the suggestion from the reviewer that this discussion is well suited to the main body, we have added the following text to Section 5.2 to make all of these points clearer:
>
> “Interestingly, the relative performance obtained from different style datasets generally differs comparably to the measurement uncertainty, which is shown for these experiments in Section A.5 of the supplementary material.  This suggests that the choice of style dataset is secondary in importance, while the main benefit comes from the use of SASSL itself.”
>
> >Do the authors have intuition as to why in certain cases, adding style from the target dataset does not achieve high performance?
>
> There are four cases (ImageNet iNaturalist, DTD, and Retinopathy) where the target dataset is used as the style dataset.  In all cases (except for Diabetic Retinopathy, as we have covered above), we observe a statistically significant improvement over the baseline from using the target dataset as the style dataset.  We also find that the average performance of the target dataset is always within 1 standard deviation of the top reported performance for that dataset (including Retinopathy).  Along with the aforementioned additional text added to the main paper, we have added further discussion of this to the supplementary material (Section A.5).
>
> >Is there a similar effect on the blending parameter?
>
> Yes, we find that there is a similar effect from the blending parameter.  We have performed ablations on this parameter (both with and without interpolation) and find that without interpolation, all blending factors perform worse than the baseline, while blending factors drawn from Uniform[0.25, 0.75] performs the best.  With interpolation drawn from Uniform[0.1, 0.3], we find that using no blending produces a modest improvement over the baseline (+0.8p.p.), while too much blending causes a performance degradation.  The drop off in performance can be intuitively understood as the input semantic information being removed from the image when the blending is too, as is shown in Figure 2.

---

### Author Response · Authors · 2024-10-30

We sincerely thank the reviewers, editors-in-chief, and action editor for their insightful comments and positive decision. Their constructive feedback has significantly improved our work.

---

### Decision · Action_Editor_PE7u · 2024-10-18

**Recommendation:** Accept as is

**Comment:**

This paper introduces a data augmentation method for self-supervised learning based on neural style transfer. The stylization is grounded in an autoencoding approach, which maintains efficiency while preserving semantic details through interpolation and blending. Experiments are conducted for image classification on ImageNet and for transfer learning.

The paper initially received positive feedback. The reviewers appreciated the idea of decoupling semantic from style information for data augmentation in self-supervised learning and found the experiments convincing. However, they raised concerns regarding the setting of hyperparameters and the impact of the style dataset on final performance. The authors' feedback was delayed.A fter the discussion period, reviewers Rw3Rd and Rw3Rd recommended acceptance, whereas RpV6L was inclined to reject the paper, mainly because they did not receive a response before making their recommendation.

The AE has carefully reviewed the submission and the discussions. The AE considers that the submission constitutes an interesting contribution to self-supervised learning, as style augmentation can complement the standard augmentations used in the literature. Overall, the authors' rebuttal, although submitted after the discussion period, was convincing regarding hyperparameter settings, training throughput, and the inclusion of new baselines, e.g., SwAV. The discussion on the choice of style datasets was more superficial but will likely inspire future work in this direction. Therefore, the AE recommends acceptance.

**Audience:**

The paper addresses data augmentation in self-supervised learning, an important topic in machine learning, which is likely to attract the interest of a broad TMLR audience.

**Claims And Evidence:**

The claims are convincingly supported by evidence, as experiments demonstrate consistent improvements over self-supervised learning baselines.